# Role of Leukotriene B_4_ Receptor-2 in Mast Cells in Allergic Airway Inflammation

**DOI:** 10.3390/ijms20122897

**Published:** 2019-06-14

**Authors:** Sun-Young Kwon, Jae-Hong Kim

**Affiliations:** Department of Biotechnology, College of Life Sciences, Korea University, 5-1 Anam-dong, Sungbuk-gu, Seoul 02841, Korea; kwonsy1992@naver.com

**Keywords:** allergic airway inflammation, mast cells, leukotriene B_4_ receptor-2, asthma

## Abstract

Mast cells are effector cells in the immune system that play an important role in the allergic airway inflammation. Recently, it was reported that BLT2, a low-affinity leukotriene (LT) B_4_ receptor, plays a pivotal role in the pathogenesis of allergic airway inflammation through its action in mast cells. We observed that highly elevated expression levels of BLT2 are critical for the pathogenesis leading to allergic airway inflammation, and that if BLT2 expression is downregulated by siBLT2-mediated knockdown, allergic inflammation is dramatically alleviated. Furthermore, we demonstrated that BLT2 mediates the synthesis of vascular endothelial growth factor (VEGF) and Th2 cytokines, such as interleukin (IL)-13, in mast cells during allergic inflammation. Based on the critical roles of BLT2 in mast cells in allergic inflammation, anti-BLT2 strategies could contribute to the development of new therapies for allergic airway inflammation.

## 1. Introduction

Mast cells are multifunctional cells in the immune system that were first identified in 1878 by Paul Ehrlich, who was awarded the Nobel Prize in Physiology or Medicine in 1908 [1,2]. Mast cells, which are key components of both innate and adaptive immunity, are involved in many biological responses, such as immune response and inflammation reactions [1,3]. Mast cells are also well known for their roles in allergic diseases, such as asthma, allergic rhinitis, atopic dermatitis, and allergic conjunctivitis [4,5,6]. In particular, mast cells play roles in the pathophysiology of the allergic airway inflammation in asthma [7,8]. In addition, previous studies suggested that mast cells play a role in neoplastic angiogenesis through the expression of vascular endothelial growth factors (VEGFs) and their receptors [9,10]. Indeed, human mast cells were shown to be both a source and a target of angiogenic and lymphangiogenic factors [9,10]. Further, mast cells were reported to participate in systemic autoimmune disorders (e.g., rheumatoid arthritis) and cancer [11,12]. Mast cells express a wide variety of membrane receptors on their cell surfaces, including high-affinity receptor for IgE (FcεRI), Toll-like receptors (TLRs), and complement receptors [13]. The cross-linking of allergen-specific IgE immune complexes and FcεRI leads to the rapid release of inflammatory mediators and cytokines (e.g., histamine, leukotrienes (LTs), and interleukin (IL)-13) by mast cells [14].

Arachidonic acid is metabolized to biologically active mediators like, thromboxanes, prostaglandins, thromboxanes, and leukotrienes (LTs) [15]. Recent work has demonstrated the importance of LTs in the pathogenesis of allergic inflammation. The LTs are formed by transformation of arachidonic acid into an unstable epoxide intermediate, LTA_4_, which can be converted enzymatically by hydration to LTB_4_, and by addition of glutathione to LTC_4_, a member of cysteinyl LTs [15]. Among LTs, LTB_4_ is a potent proinflammatory lipid mediator derived from arachidonic acid by the action of 5-lipoxygenase (5-LO) [16]. LTB_4_ is one of the most potent known chemoattractants, acting primarily on neutrophils, eosinophils, T cells, and mast cells [5,16,17,18]. LTB_4_ has been widely implicated in the pathogenesis of several inflammatory diseases, including asthma, psoriasis, rheumatoid arthritis, and inflammatory bowel disease [19]. LTB_4_ also exerts its biological functions via two types of G protein-coupled receptors (GPCRs): BLT1, which is predominantly expressed in peripheral blood leukocytes, and BLT2, which is expressed ubiquitously [19]. Previous studies have reported that both BLT1 and BLT2 are expressed in human and murine mast cells, and that both receptors contribute to the chemotactic migration of mast cells toward LTB_4_ [18,19]. LTB_4_ activates the BLT1 and BLT2, whereas additional lipid mediators (12(*S*)-hydroxyeicosatetraenoic acid (HETE) and 12-hydroxyheptadecatrienoic acid (12-HHT)) are also agonists to the BLT2 [18,19]. Previous studies have suggested that BLT2 plays a mediatory role in the pathogenesis of the allergic airway inflammation in asthma through its action in mast cells [20,21,22,23]. This review focuses on the recently discovered novel roles of BLT2 in mast cells during allergic airway inflammation.

## 2. Role of BLT2 in Mast Cells

### 2.1. Role of BLT2 in Allergen-Induced Th2 Cytokine Synthesis in Mast Cells

Mast cells are one of the major effector cells in allergic asthma and other allergic inflammatory diseases [24,25]. Mast cell-derived mediators are critical for the initiation of inflammatory responses and are associated with allergic diseases [20]. Mast cells express the high-affinity IgE receptor (FcεRI) on their surface. The cross-linking of cell surface-bound IgE to FcεRI by an allergen (Ag) leads to the rapid release of inflammatory mediators, including histamines, leukotrienes, and cytokines [24,25]. Among these mediators, Th2 cytokines (e.g., IL-4 and IL-13) produced by mast cells have been reported to regulate important allergic inflammatory responses, such as asthma, atopic dermatitis, allergic rhinitis, and anaphylaxis, as well as allergies to drugs, toxins, and food [26,27,28]. Previous studies have reported that the activation of mast cells through FcεRI leads to the production of intracellular reactive oxygen species (ROS) [29], which plays roles in the pathogenesis of various inflammatory diseases, including asthma and cancer [30,31]. Our previous study showed that BLT2 mediates the synthesis of Th2 cytokines through ROS generation in Ag-stimulated mast cells [20]. In addition, BLT2 blockade by siBLT2-mediated knockdown or antagonist treatment suppressed IL-4 and IL-13 production in allergic asthma [20]. Quite importantly, the protein levels of IL-4 and IL-13 were also modestly increased by BLT2 overexpression alone; the addition of BLT2 ligands further enhanced cytokine synthesis [20], suggesting a mediatory role of BLT2 in the allergen-induced Th2 cytokine synthesis. In contrast to BLT2, BLT1 overexpression had no significant effect on cytokine production in bone marrow-derived mast cells (BMMC) (data not shown). In summary, BLT2 may play roles in the pathogenesis of the allergic airway inflammatory response and Ag-stimulated Th2 cytokine synthesis in mast cells. In addition to the Ag–IgE action to FcεRI, TLR4 activation by lipopolysaccharide (LPS) or ST2 activation by IL-33 were also reported to stimulate the release of BLT2 agonist lipid mediators (e.g., LTB_4_ and 12(S)-HETE) that can activate BLT2, thus leading to the synthesis of Th2 cytokines, such as IL-13, as will be discussed later.

### 2.2. Role of BLT2 in Ag-Induced VEGF Synthesis in Mast Cells

Ag-induced cross-linking of cell surface-bound IgE to FcεRI activates mast cells to release the following three classes of mediators: (1) preformed mediators (e.g., histamines and proteases); (2) lipid mediators (e.g., prostaglandin D_2_ (PGD_2_), LTB_4_, and LTC_4_); and (3) a variety of chemokines and cytokines as well as VEGF [9,10,32]. VEGF is one of the regulators of vascular angiogenesis, permeability, and remodeling [33,34,35]. Angiogenesis, the process by which new capillaries develop from the preexisting vasculature, plays a key role in various pathogenesis [9]. And, mast cells were shown to express VEGFs and their receptors to mediate vascular and neoplastic angiogenesis [9,10]. A previous study reported that ROS regulate the activity of nuclear factor kappa-light-chain-enhancer of activated B cells (NF-κB) [36], a primary transcription factor involved in VEGF expression in mast cells [37,38]. In addition, mast cells regulate airway inflammation and hyperresponsiveness through orchestrating VEGF expression in allergic asthmatic responses [39]. Thus, VEGF expression in mast cells is likely to play an essential role in the initiation and development of allergic asthmatic responses. Indeed, levels of angiogenic factors, including VEGF-A, were shown to be increased in the airways of asthmatic subjects [10,40,41]. In a previous study, we observed that blocking BLT2 completely abrogated the production of VEGF in Ag-stimulated bone marrow-derived mast cells (BMMCs) [21]. The synthesis of BLT2 ligands (e.g., LTB_4_ and 12(*S*)-HETE) was also required for the VEGF production, suggesting a mediating role of an autocrine BLT2 ligands–BLT2 axis in the production of VEGF in mast cells [21]. Further, we demonstrated that the Nox1-reactive oxygen species (ROS)-NF-κB cascade is downstream of BLT2 during Ag signaling to VEGF synthesis in mast cells, implying that BLT2 mediates the synthesis of VEGF through the Nox1-ROS-NF-κB cascade in Ag-stimulated mast cells [21]. Additionally, we observed that VEGF levels in BMMCs isolated from BLT2-overexpressing TG mice were clearly elevated compared with BMMCs derived from WT mice [21]. Further, VEGF levels were much enhanced by treatment with BLT2 ligands in BMMCs derived from BLT2 TG mice [21]. In summary, these results suggest that BLT2 plays a mediatory role in VEGF production in Ag-stimulated mast cells, potentially contributing to the allergic airway inflammation.

### 2.3. Role of BLT2 in LPS-Induced IL-13 Synthesis in Mast Cells

Recently, the roles of TLRs in the pathogenesis of allergic asthma have received much attention [42]. For instance, the activation of TLR4 by lipopolysaccharide (LPS), particularly at low doses, has been reported to exacerbate allergic airway inflammation by activating mast cells and promoting Th2 responses [43,44]. In addition, it has been suggested that mast cells play a role in this response, as LPS-exacerbated airway inflammation and Th2 cytokine production are not observed in mast cell-deficient mice. Furthermore, the adoptive transfer of bone marrow-derived mast cells (BMMCs) from TLR4-deficient mice into mast cell-deficient mice does not restore allergic airway inflammation and Th2 cytokine production, while the transfer of BMMCs from wild-type mice into mast cell-deficient mice restores allergic airway inflammation [44,45]. These results suggest that TLR4 in mast cells plays a role in bacterial infection-induced exacerbation of allergic responses. Previous studies have demonstrated that Th2 cytokines induced by TLR4 activation in mast cells are critical for allergic inflammation. For example, Th2 cytokine production induced via the activation of TLR4 in mast cells by LPS was shown to be dependent on the MyD88-NF-κB signaling pathway [46,47]. Similarly, our recent study showed that BLT2 plays a pivotal role in LPS-induced Th2 cytokine production, especially IL-13 synthesis, in mast cells via a TLR4-MyD88-BLT2-ROS-NF-κB-linked cascade [22]. Moreover, we suggested that BLT2 likely acts as a potential connection between the innate and adaptive immune systems, leading to a Th2 response in mast cells [21,22].

### 2.4. Role of BLT2 in IL-33-Induced IL-13 Synthesis in Mast Cells

IL-33, a member of the IL-1 cytokine family, acts as an alarmin that is immediately released in response to epithelial cell damage [48,49,50]. Previously, IL-33 was suggested to be closely associated with asthmatic development and exacerbation in animal model study, as IL-33 levels are increased when the severity of asthma is increased in mouse [51]. The relationship between IL-33 and asthma has been demonstrated using an IL-33-overexpressing transgenic mouse, which showed asthmatic airway inflammation with an influx of inflammatory cells and increased expression levels of Th2 cytokines [52]. Consistent with the proposed role of IL-33 in asthma, earlier studies showed that blockade of IL-33 or its cell surface receptor ST2 clearly reduces disease severity, as evidenced by reduced Th2 cytokine production, airway inflammation, airway hyperresponsiveness, and mucus production [53]. Although mast cells are known to respond to exogenous danger signals [50], recent studies have suggested that mast cells play a critical role in the recognition of endogenous danger signals such as IL-33 [50]. Indeed, several studies have demonstrated that mast cells are a major ST2-expressing cell type and are associated with the IL-33-related development of asthma [53,54,55]. For example, the activation of mast cells by IL-33 has been shown to enhance airway hyperresponsiveness and asthmatic inflammation via the production of Th2 cytokines such as IL-13 [48,56,57]. In addition, IL-13 secreted from IL-33-stimulated mast cells was shown to increase airway smooth muscle contraction and contribute to the development of airway inflammation [57]. Previous studies have reported that 5-LO or 12-lipoxygenase (12-LO) is closely associated with the development of allergic asthmatic inflammation in animal model studies [58,59,60]. For instance, LTB_4_, a 5-LO product, has been demonstrated to perform important functions during the development of asthma. In consistent with these findings in animal studies, HETE, a 12-LO product, was detected at higher levels in the sputum and bronchoalveolar lavage fluid (BALF) of asthmatic patients than in those of healthy subjects [59]. These findings suggest that 5-/12-LO are involved in the development of asthmatic airway inflammation in both animal model and clinical subjects. Recently, we found that 5-/12-LO plays critical roles in IL-33 signaling to induce IL-13 production in mast cells [23]. In addition, we reported that the “MyD88-5-/12-LO-BLT2-NF-κB” signaling pathway contributes to IL-33-induced IL-13 synthesis in mast cells, potentially exacerbating the airway inflammation. Thus, the action of BLT2 in mast cells plays a role in IL-33 signaling, potentially contributing to IL-13 synthesis and allergic airway inflammation. Quite recently, IL-33 was reported to participate in innate and adaptive immunity and inflammation, and acting on CD34+ cells causes mast cells differentiation and maturation [61,62,63]. Clearly, further studies are necessary to investigate the roles of BLT2 in mast cells, especially in terms of mast cells stimulation, differentiation, and maturation.

## 3. Conclusions

Mast cells, which are principal effector cells in the immune system, play roles in the allergic airway inflammation. In recent studies, we demonstrated that BLT2 as well as its ligands LTB_4_ and 12-HHT plays an important role in the synthesis of VEGF and Th2 cytokines in Ag-stimulated mast cells, contributing to allergic airway inflammation. In addition, we demonstrated that this autocrine-acting BLT2 cascade mediates LPS/TLR4- or IL-33-induced synthesis of Th2 cytokines in mast cells. Based on those findings, we proposed a hypothetical model as summarized in Figure 1. 

This model proposes that BLT2 likely acts as a potential connection between the innate and adaptive immune systems, leading to a Th2 response, at least in mast cells. In this model, NOX1/ROS and NF-κB lies downstream of BLT2, thus mediating the innate and adaptive signaling for IL-13 production. Thus, understanding the detailed roles of BLT2 in mast cells during allergic airway inflammation may contribute to the development of new therapeutic targets for allergic airway inflammation. Indeed, we are currently working on the development of BLT2 receptor antagonists as a novel therapy for allergic disorders.

## Figures and Tables

**Figure 1 ijms-20-02897-f001:**
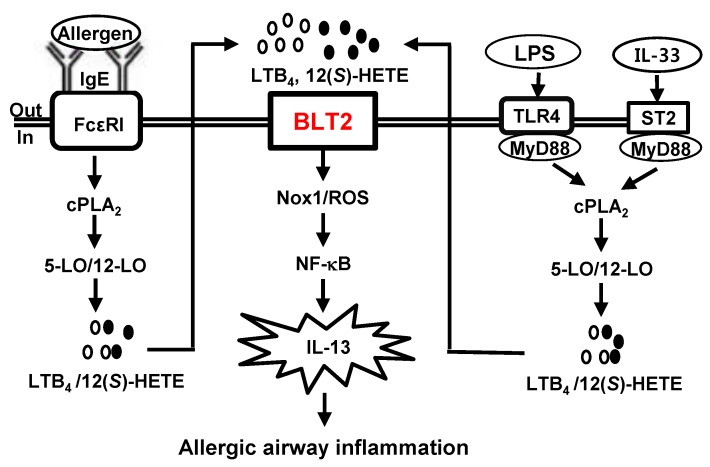
Scheme of BLT2-dependent lipopolysaccharide/Toll-like receptors 4 (LPS/TLR4), interleukin (IL)-33/ST2, or allergen/ high-affinity IgE receptor (FcεRI) immune response signaling pathways responsible for IL-13 synthesis in mast cells, contributing to the allergic airway inflammation. Mast cells activation by allergen/FcεRI, LPS/TLR4, or IL-33/ST2 stimuli activates cytosolic phospholipase A_2_(cPLA_2_)-5-/12-LO-BLT2 cascades and induces allergic airway inflammation by significantly increasing Th2 cytokines synthesis (e.g., IL-13) production via the reactive oxygen species-nuclear factor kappa-light-chain-enhancer of activated B cells (ROS-NFκB) pathway.

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
