# Peer review of "Role of Leukotriene B4 Receptor-2 in Mast Cells in Allergic Airway Inflammation"

_ijms, 2019, doi:10.3390/ijms20122897_

Reviewer 1 Report

Title: Role of leukotriene B4 receptor-2 in mast cells in allergic airway inflammation.

 -In this article the authors study LTB4 in the pathogenesis of allergic airway inflammation in relation to mast cells. The authors find that a LTB4 BLT2 receptor is elevated and is fundamental for inflammatory pathogenesis. They conclude that anti-BLT2 strategies could contribute to the development of new therapies for allergic airway inflammation.

 - This short paper is good but it should be improved by addressing the topic more widely and not just limited to the result obtained. So I have some concerns that should be clarified.

 -This paper should be expanded. When the authors talk about allergy, respiratory infections and modulation of the immune system must be considered. Some dietary supplements can help prevent and reduce allergic inflammation. Therefore, to make this paper more complete and interesting for the readers of this important journal, the authors should expand a bit the discussion on this subject.

Below I list an interesting article that has recently been reported and should be studied, the meaning incorporated and reported briefly in the discussion and in the list of references.

Ameli F, Ciprandi G. Sinerga may prevent recurrent respiratory infections in allergic children. J Biol Regul Homeost Agents. 2019 Mar-Apr;33(2):601-607.

 - The authors also talk about the stimulation and inhibition of mast cells, but this argument is not sufficiently and clearly described. Below two new interesting articles can improve this theme. Then, again, I suggest that these articles should be studied, incorporate the meaning and report briefly in the discussion and in the reference list.

Gallenga CE, Pandolfi F, Caraffa A, et al. Interleukin-1 family cytokines and mast cells: activation and inhibition. J Biol Regul Homeost Agents. 2019 Jan-Feb,;33(1):1-6.

Varvara G, Tettamanti L, Gallenga CE, et al. Stimulated mast cells release inflammatory cytokines: potential suppression and therapeutical aspects. J Biol Regul Homeost Agents. 2018 Nov-Dec;32(6):1355-1360.

 -To better address the theme of IL-33, here too I have suggestions. IL-33 participates in innate and adaptive immunity and inflammation and, acting on CD34+ cells, causes MC differentiation and maturation. This theme is very well addressed in this recent article below. Therefore, again, I suggest that these articles should be studied, incorporate the meaning and report briefly in the discussion and in the reference list.

Tettamanti L, Kritas SK, Gallenga CE, et al. IL-33 mediates allergy through mast cell activation: Potential inhibitory effect of certain cytokines. J Biol Regul Homeost Agents. 2018 Sep-Oct;32(5):1061-1065.

 -Figure 1 has to be redone because in the stimulation with the allergen, the biochemical cascade, leading to LTB4, is too simple and skips some steps that eventually lead to the formation of LTB4.

Furthermore, the stimulation with IL-33 seems not to occur on the cell, but on LTB4.

LPS activation also lacks some biochemical steps and the TLR-supplied cell should be inserted.

The legend says very little and needs to be expanded.

 -I believe these suggestions are important for improving this paper. Without these corrections the paper cannot be published. So I recommend minor revision.

-I'd like to review this article after corrections.

Title: Role of leukotriene B4 receptor-2 in mast cells in allergic airway inflammation.

 -In this article the authors study LTB4 in the pathogenesis of allergic airway inflammation in relation to mast cells. The authors find that a LTB4 BLT2 receptor is elevated and is fundamental for inflammatory pathogenesis. They conclude that anti-BLT2 strategies could contribute to the development of new therapies for allergic airway inflammation.

 - This short paper is good but it should be improved by addressing the topic more widely and not just limited to the result obtained. So I have some concerns that should be clarified.

 -This paper should be expanded. When the authors talk about allergy, respiratory infections and modulation of the immune system must be considered. Some dietary supplements can help prevent and reduce allergic inflammation. Therefore, to make this paper more complete and interesting for the readers of this important journal, the authors should expand a bit the discussion on this subject.

Below I list an interesting article that has recently been reported and should be studied, the meaning incorporated and reported briefly in the discussion and in the list of references.

Ameli F, Ciprandi G. Sinerga may prevent recurrent respiratory infections in allergic children. J Biol Regul Homeost Agents. 2019 Mar-Apr;33(2):601-607.

 - The authors also talk about the stimulation and inhibition of mast cells, but this argument is not sufficiently and clearly described. Below two new interesting articles can improve this theme. Then, again, I suggest that these articles should be studied, incorporate the meaning and report briefly in the discussion and in the reference list.

Gallenga CE, Pandolfi F, Caraffa A, et al. Interleukin-1 family cytokines and mast cells: activation and inhibition. J Biol Regul Homeost Agents. 2019 Jan-Feb,;33(1):1-6.

Varvara G, Tettamanti L, Gallenga CE, et al. Stimulated mast cells release inflammatory cytokines: potential suppression and therapeutical aspects. J Biol Regul Homeost Agents. 2018 Nov-Dec;32(6):1355-1360.

 -To better address the theme of IL-33, here too I have suggestions. IL-33 participates in innate and adaptive immunity and inflammation and, acting on CD34+ cells, causes MC differentiation and maturation. This theme is very well addressed in this recent article below. Therefore, again, I suggest that these articles should be studied, incorporate the meaning and report briefly in the discussion and in the reference list.

Tettamanti L, Kritas SK, Gallenga CE, et al. IL-33 mediates allergy through mast cell activation: Potential inhibitory effect of certain cytokines. J Biol Regul Homeost Agents. 2018 Sep-Oct;32(5):1061-1065.

 -Figure 1 has to be redone because in the stimulation with the allergen, the biochemical cascade, leading to LTB4, is too simple and skips some steps that eventually lead to the formation of LTB4.

Furthermore, the stimulation with IL-33 seems not to occur on the cell, but on LTB4.

LPS activation also lacks some biochemical steps and the TLR-supplied cell should be inserted.

The legend says very little and needs to be expanded.

 -I believe these suggestions are important for improving this paper. Without these corrections the paper cannot be published. So I recommend minor revision.

-I'd like to review this article after corrections.

Author Response

Date: 2019-06-11

 Manuscript Number: IJMS-527374 (Opinion)

Title: Role of leukotriene B4 receptor-2 in mast cells in allergic airway inflammation

 Dear Reviewer 1

Thank you very much for the valuable comments regarding our manuscript entitled “Role of leukotriene B4 receptor-2 in mast cells in allergic airway inflammation". As you suggested, I revised the manuscript. I hope this revision will satisfy many issues raised by you. I am confident that the revised manuscript more clearly propose the novel role of BLT2 in mast cells in allergic airway inflammation.

 The following issues were revised in the manuscript.

 1. As the reviewer suggested, I tried to expand the theme of the manuscript to include the recent findings on the stimulation/inhibition of mast cells as well as IL-33 (see Line 159~). Also, the references are included in the list 61-63.

2. Figure 1 was redone and also the legend was expaned (see Figure 1 and the legend).

 Thank you again for your suggestions on our manuscript, and I look forward to hearing from you soon.

 Sincerely yours,

Jae-Hong Kim, Ph.D

Department of Biotechnology

College of Life Sciences and Biotechnology

Korea University

5-1 Anam-dong, Sungbuk-gu

Seoul, 136-701, Korea

 Reviewer 2 Report

The manuscript by drs. Kwon and Kim provides an interesting review of the role of the BLT2 receptor in allergic airway inflammation. Agonists that are able to activate the BLT2 receptor on mast cells are generated via three pathways: via allergens that crosslink the high affinity IgE-receptor, by LPS-TLR4 and by IL33 acting on the ST2 receptor. The resulting BLT2 agonists trigger reactive oxygen species, which activate NFkB to enhance transcription of the type 2 cytokine IL13.

I have a series of general comments on the writing style, and a series of questions about the content of the manuscript (all with the intention to improve the quality of the manuscript further). 

In the first place, I would like to ask the authors to minimize the use of qualifying words like “important, critical, dramatic, major, central, essential or pivotal”. Currently almost every second sentence uses one of these “advertisement” words. Just give an emotion-free account the data! The other aspect that is somewhat annoying is the over-use of the word “recent”. Checking these “recent discoveries” in the reference list shows that these are by no means recent, but rather refer to articles at least two years ago, and in one case from 2005 (citation #47). A third point that I would propose to the authors: they give a statement, and then provide the data. I would prefer if they provide the data, and then give their conclusion or interpretation. Examples are: Line 24 “Mast cells ..... are involved in many responses” (such as???). Details follow in the subsequent sentences. Or, Line 61: “Th2 cytokines ..... regulate responses in asthma” (which?). As a last general point, I would like to ask the authors to carefully state the species, as currently it is difficult to check if something has been observed in a preclinical study, or whether the observation comes from a study in patients.

Now to the details: 

The abstract mentions IL-4, but the rest of the manuscript strongly focuses on IL-13. (Leave out IL-4 from abstract, or extend the text with more IL-4 data?).  

Introduction is numbered 1. But thereafter there is no further numbering of paragraphs. Line 36: change “in mast cells” to “by mast cells”. The next sentence starting on line 38 should be reformulated: Arachidonic acid is metabolized to biologically active mediators like, thromboxane, prostagandins and leukotrienes. Also the sentence on Line 46 should be reformulated: LTB4 activates the BLT1 and BLT2, whereas additional lipid mediators (12-HETE or 12HHT) are also agonists at the BLT2 receptor. [At this point it might be mentioned why these products are also important for allergic diseases]. In the introduction, the authors should also mention why BLT1 is not (or not so much) relevant for allergy.

The next paragraph (Role of BLT2). This section again starts with explaining how important mast cells are (this was already done in the introduction). I think, that this section is a good place to mention the three pathways (allegen-IgE-receptor, by LPS-TLR4 and by IL33-ST2 receptor) leading to lipid mediators that can activate the BLT2 (also expressed by mast cells), and this ending up in synthesis and release of IL-13 and VEGF. On line 63: "various inflammatory diseases, including asthma and cancer".

Role of BLT2 in VEGF synthesis paragraph. What is the pathway via which BLT2 stimulation leads to VEGF transcription? Is it IgE – FceR1 – VEGF? At what point is then BLT2 involved? Is the evidence for BLT2 based on only the BLT2 transgenic mice? This section should be extended (or, depending on the existing evidence, shortened).  

Paragraph on IL-33. IL33 levels are increased when the severity of asthma is increased. This is a correlation, but does not prove causation. The evidence for a causal role of IL33 is given in the next sentence. I would propose to reverse the rank order of the two sentences. Also, please make sure to carefully separate data from human and laboratory animals. “AHR” should be written in full. What is the pathway for IL33? Is it BLT2 – NFkB – IL33? Or does IL33 cause an increase LTB4? (This could/should be added to Figure 1.). 

Figure 1. This is a very useful picture and summarizes the content of the manuscript at one glance. I would like to see a more extended legend with more explanation. It should include a pathway for IL33. Is IL4 transcribed in parallel to IL13? 

Finally, the major message of the manuscript is: BLT2 receptor antagonists could be a useful therapy for allergic disorders. This begs the question: what is the current state of drug-development? Are there tool compounds, and do they hold up to the expectation? A few words on this topic might be helpful! 

Author Response

Date: 2019-06-11

 Manuscript Number: IJMS-527374 (Opinion)

Title: Role of leukotriene B4 receptor-2 in mast cells in allergic airway inflammation

Authors: Sun-Young Kwon, Jae-Hong Kim

 Dear Reviewer 2

 Thank you very much for the valuable comments regarding our manuscript entitled “Role of leukotriene B4 receptor-2 in mast cells in allergic airway inflammation". As you suggested, I revised the manuscript. I hope this revision will satisfy many issues raised by you. I am confident that the revised manuscript more clearly propose the novel role of BLT2 in mast cells in allergic airway inflammation.

 The following issues were revised in the manuscript.

 1. As the reviewer suggested, I tried to minimize the use of qualifying words throughout the text (see the attached mansucript PDF file).

2. As the reviewer pointed, "recent" word was edited to "previous". I am so sorry about this error.

Additionally, the many points raised by the reviewer were revised in the manuscript. For example,

3. Numbering (Line 59, 83, 110, 129, 163);

4. Three pathways for mast cell activation (Line 79~);

5. VEGF pathway (Line 102~);

6. IL-33 and the species (human or animal) (Line 132, 133, 149, 151, 152, 153, 154);

7. Figure 1 and legend (see Figure 1 and its legend in the revised manuscript);

8. Current stage of BLT2-based drug development (Line 183~).

Thank you again for your suggestions on our manuscript, and I look forward to hearing from you soon.

 Sincerely yours,

 Jae-Hong Kim, Ph.D

Department of Biotechnology

College of Life Sciences and Biotechnology

Korea University

5-1 Anam-dong, Sungbuk-gu

Seoul, 136-701, Korea

Round  2

Reviewer 1 Report

As the Authors have carried out all the changes I suggested, I believe that this paper can now be accepted.